# Hemidesmosomes and Notch signaling regulate epidermal differentiation via delamination

Juliet S. King[1], Kendall J. Lough[1,2] and Scott E. Williams[1],*

## ABSTRACT

Integrins mediate adhesion of basal keratinocytes to the underlying basement membrane. While high expression of integrins has been correlated with stemness, there is limited direct evidence that integrins mediate keratinocyte retention within the basal layer. Here, we generate mosaic, epidermal-specific loss of integrin-β4 (encoded by *Itgb4*) or its ligand, laminin-α3β3γ2 (*Lama3*), in mouse using an *in utero* lentiviral-mediated approach. Although mutations in these genes cause postnatal skin blistering in mice and humans, we observe no evidence of dermal-epidermal separation embryonically. Despite no obvious alterations to apicobasal polarity, *Itgb4*-deficient basal cells show mild defects in oriented cell divisions, with increased oblique divisions and altered telophase correction. However, differentiation via delamination, whereby basal keratinocytes lose adhesion to the underlying basement membrane and transit into the suprabasal layer, is elevated upon *Itgb4* or *Lama3* loss. Notably, hyperactive Notch signaling both decreases integrin-β4 expression and increases delamination, while deletion of the Notch effector *Rbpj* has the opposite effect. These findings conclusively demonstrate a causal role for hemidesmosomes in regulating epidermal differentiation through both mitotic and non-mitotic mechanisms, and shed additional light on the programs regulating delamination.

KEY WORDS: Hemidesmosome, Delamination, Integrin, Differentiation, Notch, Epidermis, Mouse

## INTRODUCTION

The epidermis is a stratified epithelium consisting of a proliferative basal layer overlaid with suprabasal layers that differentiate via either asymmetric cell divisions (ACDs) or delamination. In an ACD, the spindle aligns along the apicobasal axis, displacing one daughter suprabasally, whereas during delamination cells exit the basal layer without dividing. Both ACDs and delamination contribute to epidermal differentiation during embryogenesis, with delamination driving initial stratification and ACDs peaking later (Damen et al., 2021; Descovich et al., 2023; Lough et al., 2020; Williams et al., 2014). In adulthood, ACDs are uncommon and

differentiation is driven mainly by delamination (Cockburn et al., 2022; Ipponjima et al., 2016; Liu et al., 2019; Rompolas et al., 2016). Most evidence of delamination has been indirect, and little is known about what regulates this process *in vivo* (Cockburn et al., 2022; Damen et al., 2021; Ellis et al., 2019; Mesa et al., 2018; Miroshnikova et al., 2018).

Basal keratinocytes adhere to an underlying extracellular matrix-rich basement membrane via two major classes of integrins. Integrin-α6β4 dimers form a complex with the cytokeratin intermediate filament network to form hemidesmosomes (HDs), whereas α3β1 dimers integrate into focal adhesions and link to the actin cytoskeleton (Rousselle et al., 2022). The importance of HDs is exemplified by the severe blistering disease that occurs in mice and humans upon disruption of HD components or their ligand, laminin-α3β3γ2 (laminin-332), known as junctional epidermolysis bullosa (JEB) (Dowling et al., 1996; Pulkkinen and Uitto, 1998; Raymond et al., 2005). High levels of integrin-β1 have been linked to keratinocyte stemness, while loss of surface integrin-β1 and integrin-α6 are correlated with differentiation (Adams and Watt, 1990; Jones et al., 1995; Li et al., 1998). Paradoxically, expression of differentiation markers precedes loss of surface integrins (Adams and Watt, 1990; Cockburn et al., 2022), raising the question of whether integrin loss is a cause or consequence of differentiation. Adding to this complexity, there is no evidence that integrin loss impacts epidermal differentiation, even in non-blistered and embryonic skin (DiPersio et al., 2000; Dowling et al., 1996; Georges-Labouesse et al., 1996; van der Neut et al., 1996).

Here, we analyze the consequences of mosaic *Itgb4* or *Lama3* loss in the developing epidermis and show that mutant cells are more prone to differentiate than their wild-type (WT) counterparts. While *Itgb4* loss has only subtle effects on division orientation, delamination is markedly increased upon loss of *Itgb4* or its ligand *Lama3*. Finally, Notch signaling, an epidermal differentiation regulator, inhibits integrin-β4 expression and promotes delamination. Collectively, these studies demonstrate a role for HDs in promoting basal layer retention and provide mechanistic insights into the pathways that regulate delamination.

## RESULTS AND DISCUSSION

Since germline or whole epidermal loss of *Itgb4* results in dermal-epidermal separation (Dowling et al., 1996; Raymond et al., 2005), we utilized an *in utero* lentiviral-mediated approach to generate mosaic *Itgb4* knockdown (Fig. 1A). Using this LUGGIGE (lentiviral ultrasound-guided gene expression and gene inactivation) technique (Beronja et al., 2010), basal keratinocytes were transduced at embryonic day (E) 9.5, prior to stratification, with nuclear RFP signaling viral infection. RFP+ cell phenotypes can then be compared to both uninfected littermates (WT) and neighboring, uninfected cells (RFP−). *Itgb4*^4124 and *Itgb4*^2326 RFP+ cells showed robust depletion of *Itgb4* at the mRNA level *in vitro*,

[1]Departments of Pathology & Laboratory Medicine and Biology, Lineberger Comprehensive Cancer Center, The University of North Carolina, Chapel Hill, NC 27959, USA. [2]Center for Gastrointestinal Biology and Disease, Bowles Center for Alcohol Studies, The University of North Carolina, Chapel Hill, NC 27959, USA.

*Author for correspondence (scott_williams@med.unc.edu)

J.S.K., 0000-0003-3865-6956; S.E.W., 0000-0001-9975-7334

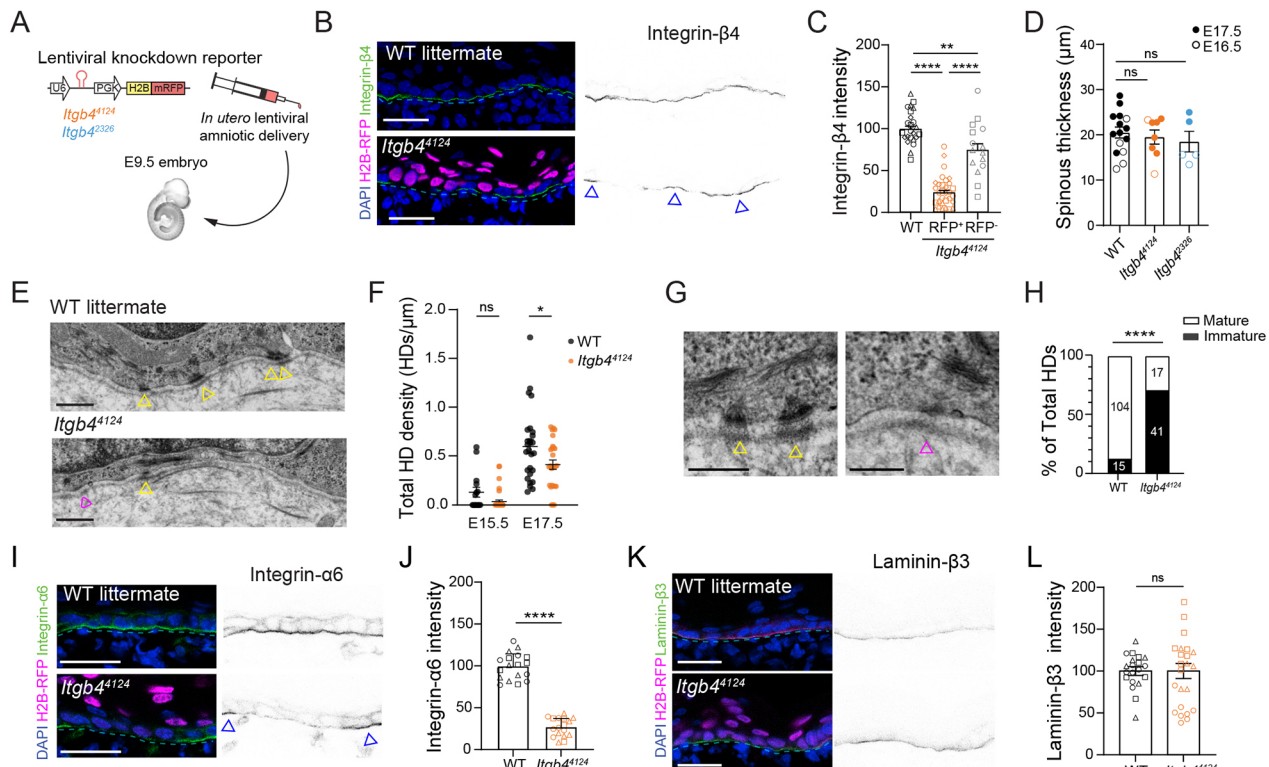

**Fig. 1. Mosaic loss of *Itgb4* alters embryonic basement membrane organization without overt blistering.** (A) Schematic of the LUGGIGE technique. (B,C) Single-plane confocal images of E17.5 mosaic epidermis showing integrin-β4 reduction beneath *Itgb4⁴¹²⁴* RFP⁺ knockdown cells (B), and associated quantification from *n*≥6 embryos per condition; independent images from each litter are represented by a unique shape (triangle, circle or square) (C). (D) Spinous thickness determined by quantifying cytokeratin 10 (K10) immunofluorescence in E16.5-E17.5 control (WT), *Itgb4⁴¹²⁴* and *Itgb4²³²⁶* epidermis; each dot in represents average thickness per animal. (E-H) Transmission electron micrographs (E,G) and quantification (F,H) of hemidesmosomes (HDs) in E17.5 control and highly transduced *Itgb4⁴¹²⁴* nape skin. Total HD density is quantified in H and maturity in J; yellow arrowheads indicate mature and magenta arrowheads immature HDs. (I-L) Single-plane confocal images of E17.5 *Itgb4⁴¹²⁴* knockdown epidermis showing integrin-α6 (I) and laminin-β3 (K) with intensity quantification (J,L). Basement membrane is indicated with cyan dashed line; blue arrowheads (B,I) represent areas of non-transduced (RFP⁻) basal cells. Scale bars: 25 μm (B,I,K); 0.5 μm (E); 0.25 μm (G). ns, not significant; *P<0.05, **P<0.01, ****P<0.0001 (unpaired *t*-test or Mann-Whitney test).

and of integrin-β4 protein at the dermal-epidermal junction (DEJ) *in vivo* (Fig. 1B-D; Fig. S1A-C).

*Itgb4⁴¹²⁴* and *Itgb4²³²⁶* epidermis had equivalent thickness of the suprabasal [K10 (Krt10, keratin 10)⁺] layers compared to littermates (Fig. 1D; Fig. S1D). There were no overt signs of blistering, so we utilized transmission electron microscopy (TEM) to examine the DEJ ultrastructure. HDs were sparse at E15.5 in both the *Itgb4⁴¹²⁴* and age-matched WT DEJ. At E17.5, in agreement with previous neonatal studies (Dowling et al., 1996; Raymond et al., 2005; van der Neut et al., 1996), *Itgb4⁴¹²⁴* epidermis showed reduced HD density compared to WT controls in highly transduced nape skin (Fig. 1E,F). HD morphology was also variable, with *Itgb4⁴¹²⁴* tissue having significantly fewer mature, electron-dense HDs (Fig. 1G,H; Fig. S1E,F). A significant reduction in hemidesmosomal integrin-α6 was also observed in RFP⁺ regions of *Itgb4⁴¹²⁴* epidermis, while neither the focal adhesion protein integrin-β1 nor the basement membrane component laminin-β3 appeared to be grossly affected by *Itgb4* loss (Fig. 1I-L; Fig. SG,H).

Although *Itgb4* knockdown epidermis showed normal expression of differentiation markers, we noticed that RFP⁺ cells appeared more abundant in suprabasal layers, suggesting an increased propensity to differentiate (Fig. 2A). We reasoned that all RFP⁺ spinous cells are the result of either an asymmetric division or delamination event. In this way, RFP positivity serves as a lineage trace, and is particularly effective when transduction levels are at

low 'clonal' density. We defined the 'differentiation index' as the proportion of suprabasal cells that are RFP⁺ divided by the proportion of basal cells that are RFP⁺. 'Scramble' controls had a differentiation index near one (1.13), while *Itgb4²³²⁶* and *Itgb4⁴¹²⁴* epidermis favored suprabasal enrichment, with ratios skewing higher (2.61 and 1.73, respectively; Fig. 2B-D). Therefore, integrin-β4 loss leads to a comparative disadvantage in basal layer occupancy, favoring differentiation.

Hypoproliferation could account for reduced numbers of basal RFP⁺ cells, but we did not observe differences in either the mitotic marker phospho-histone H3 (pHH3) or incorporation of 5-ethynyl-2-deoxyuridine (EdU) (Fig. S2A-D). Although apoptotic cells were rare, there was no evidence of elevated cell death upon *Itgb4* knockdown (Fig. S2E,F). Next, we investigated whether a shift in division orientation could explain the basal occupancy deficit. During peak stratification (E16.5-E17.5), there are roughly equal proportions of planar symmetric cell divisions and perpendicular differentiative ACDs, with very few oblique-oriented divisions (Lough et al., 2019). Using the mid-body marker survivin (BIRC5), we measured the division angle of telophase cells relative to the basement membrane in RFP⁺, RFP⁻ and WT littermate basal cells (Fig. 2E). In both *Itgb4⁴¹²⁴* and *Itgb4²³²⁶* epidermis, we observed an atypical, but insignificant increase in oblique divisions at the expense of planar divisions (Fig. 2F,G; Fig. S3A). Importantly, *Itgb4⁴¹²⁴* cells maintained proper apicobasal polarity and, similar to germline-deleted *Itgb4* epidermis

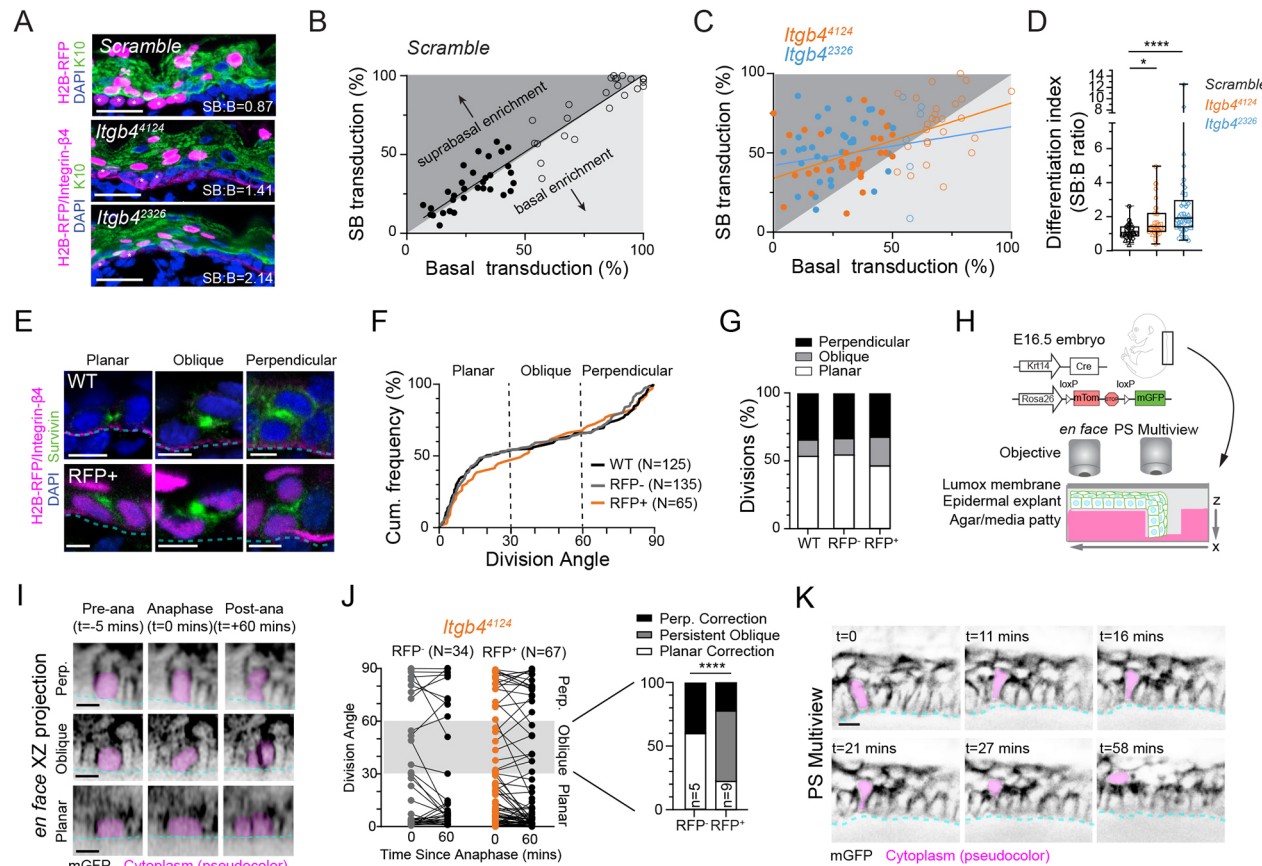

**Fig. 2. *Itgb4* loss leads to decreased basal occupancy but does not impact division orientation or polarity.** (A) Confocal images of Scramble control and *Itgb4* knockdown epidermis showing RFP⁺ relative to K10; differentiation index (SB:B transduction ratio) is shown for each image. (B,C) Plots of suprabasal versus basal transduction for scramble controls (B) and *Itgb4⁴¹²⁴* and *Itgb4²³²⁶* (C) epidermis; each dot indicates one field of view (FOV) with solid circles <50% basal transduction and empty circles >50%; data collected from *n*≥4 biological replicates per condition. (D) Quantification of differentiation indices (SB:B ratio) for FOVs with <50% basal transduction; each shape represents a unique litter; whiskers represent min/max; box represents 25th-75th percentile with center line as median. (E) WT and *Itgb4⁴¹²⁴* RFP⁺ telophase cells (survivin, green) in planar, oblique and perpendicular orientations. (F) Cumulative frequency plot of division angles for WT (black) and *Itgb4⁴¹²⁴* RFP⁺ (orange) and RFP⁻ (gray) cells; no significant differences between groups by Kolmogorov–Smirnov cumulative frequency test. (G) Bar graph of orientation types, binned by 30° increments. (H) Schematic of *ex vivo* live imaging with *en face* and PS-Multiview settings. (I) *xz* projections of dividing cells at pre-anaphase (t=−5 min), anaphase (t=0 min) and post-anaphase (t=60 min) time points. (J) Line graphs of division orientation angle per cell between anaphase and 60 min post-anaphase in *Itgb4⁴¹²⁴* RFP⁻ (gray) and RFP⁺ cells (orange) with bar graph depicting outcomes of initial oblique cells at right. (K) PS-Multiview timelapse of a WT cell (pink) undergoing delamination. Basement membrane is indicated with cyan dashed lines. Scale bars: 10 μm (E,I,K); 25 μm (A). ns, not significant; *P<0.05, ****P<0.0001 [Mann-Whitney test (D), Kolmogorov-Smirnov test (F), Chi-squared test (G,J)].

(Lechler and Fuchs, 2005), LGN, which is required for perpendicular divisions (Williams et al., 2011), showed normal apical localization in mitotic *Itgb4⁴¹²⁴* mitotic cells. (Fig. S3B-G). This suggests that the mild increase in oblique divisions cannot be explained by defects in initial spindle orientation.

Alternatively, the mild increase in oblique divisions could be explained by altered telophase correction, if displaced apical daughters failed to properly reintegrate into the basal layer (Lough et al., 2019). To test this, we performed *en face ex vivo* live imaging (Cetera et al., 2018; Descovich et al., 2023; Lough et al., 2019) (Fig. 2H) on mosaically transduced *Itgb4⁴¹²⁴* E16.5 *Krt14^Cre^; Rosa26^mTmG^* epidermal explants, and measured the division angle from anaphase onset (time, t=0) to 60 min post-anaphase (t=60; Fig. 2I). Generally, both RFP⁻ and RFP⁺ cells that entered anaphase at planar or perpendicular orientations remained in these positions through telophase. However, telophase obliques were more frequent among RFP⁺ cells (12%) compared to RFP⁻ cells (3%), and the majority (55%) of RFP⁺ cells that entered anaphase at an oblique angle remained obliquely positioned, a behavior never observed in RFP⁻

cells (Fig. 2J; example shown in Fig. S3H,I). These observations suggest that the mild increase in oblique divisions observed in fixed tissue is likely attributable to impaired telophase correction.

In addition to ACDs, basal keratinocytes differentiate through a division-independent mechanism termed delamination. Although poorly understood at a molecular level, delaminating cells initiate expression of differentiation markers while still in the basal layer (Cockburn et al., 2022; Damen et al., 2021; Williams et al., 2014). Delamination has only been directly observed in embryogenesis in 'jelly roll' explants, which, although being close to a wound edge, offer clear *z*-resolution (Damen et al., 2021). To adopt the optical advantage of *z*-plane imaging in a physiologically intact tissue, we utilized the planar-sagittal (PS)-Multiview approach (Jones et al., 2023) (Fig. 2H). This enabled us to observe rare delamination events, occurring on a time scale of 30-60 min (Fig. 2K), in agreement with previous reports (Damen et al., 2021). Of note, delaminating cells adopted a distinct morphology with a narrow basal process (Fig. 2K, second to fourth frames), consistent with progressive loss of basement membrane adhesion.

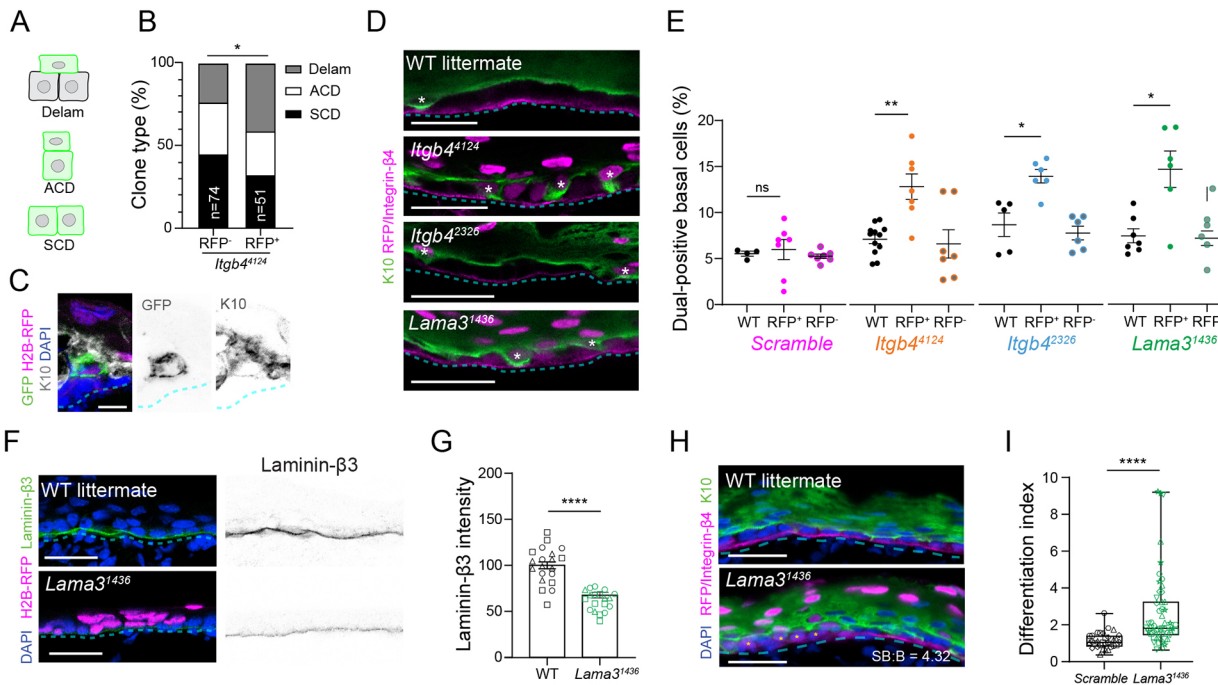

**Fig. 3. Loss of integrin-β4 or its ligand laminin-332 increases delamination.** (A-C) Clonal lineage tracing, with cartoons of clone types (A), representative image of *Itgb4⁴¹²⁴* RFP⁺ delamination clone (C), and distribution of *Itgb4⁴¹²⁴* RFP⁻ versus RFP⁺ clones (B). (D) Confocal images of K10⁺/K14⁺ basal cells in WT, *Itgb4⁴¹²⁴*, *Itgb4²³²⁶* and *Lama3¹⁴³⁶* epidermis; asterisks indicate dual-positive cells. (E) Quantification of dual-positive frequency; each dot represents the total percentage for an animal, per genotype. (F,G) Representative images of laminin-β3 intensity in control and *Lama3¹⁴³⁶* epidermis (F), with associated quantification (G). (H) Confocal images of control and *Lama3¹⁴³⁶* epidermis showing RFP⁺ cells relative to K10; asterisks indicate basal-resident RFP⁺ cells. (I) SB:B ratio (differentiation index) quantification; whiskers represent min/max; box represents 25th-75th percentile with center line as median. In G,I, each point represents a FOV; shapes indicate unique litters; *n*>3 biological replicates/condition. Basement membrane is indicated with cyan dashed lines. Scale bars: 25 μm (D,F,H); 10 μm i(C). ns, not significant; *P<0.05, **P<0.01, ****P<0.0001 [Chi-squared test, unpaired *t*-test (E,G) or Mann-Whitney test (I)].

Since genetic lineage tracing can be used to infer cellular behaviors based on clone morphology (Descovich et al., 2023; Lough et al., 2019; Williams et al., 2014), we induced small clones on an *Itgb4⁴¹²⁴* mosaic background and compared RFP⁻ (WT) and RFP⁺ (knockdown) clone types (Fig. 3A). *Itgb4⁴¹²⁴* knockdown clones showed a significant increase in delamination events compared to RFP⁻ controls (Fig. 3B,C). Notably, the balance of asymmetric and symmetric cell divisions was unaffected by *Itgb4* loss, consistent with the lack of an observed spindle orientation phenotype (Fig. 2F). Thus, integrin-β4 impacts differentiation largely by influencing delamination behavior.

Since both PS-Multiview and LUGGIGE clonal lineage tracing are relatively low throughput, we turned to fixed imaging to assess delamination rates across a variety of HD mutants. Historically, delaminating cells have been inferred based on their expression of both basal and suprabasal keratins (Krt14 and Krt10, respectively) while maintaining residual contact to the basement membrane (Williams et al., 2014; Miroshnikova et al., 2018). At E17, both short hairpin RNAs (shRNAs) targeting *Itgb4* resulted in a ~2-fold increase in dual-positive basal cells compared to RFP⁻ and WT controls (Fig. 3D,E). This phenotype was age specific, because no differences in delamination frequency were observed in the HD-poor, E15.5 epidermis (Fig. S4A).

Integrin-α6β4 dimers bind to laminin-332, which is produced and secreted by basal keratinocytes (Rousselle and Beck, 2013; Rousselle et al., 2022). In addition to decreased maturity and number of mature HDs, *Itgb4⁴¹²⁴* tissue showed loops and occasional breaks in the lamina densa (pseudocolored cyan in Fig. S4B). As an additional means to investigate the role of hemidesmosomal adhesions in epidermal differentiation, we designed a shRNA, *Lama3¹⁴³⁶*, to target

one of the laminin-332 subunits. As validation of its efficacy *in vivo*, we observed a 35% reduction of laminin-β3 intensity at the DEJ, indicating successful reduction of the laminin-332 trimer (Fig. 3F,G). Like *Itgb4*, *Lama3* loss did not affect epidermal thickness (Fig. S4C). However, *Lama3* loss led to a significant increase in the differentiation index (2.69, *P*<0.0001; Fig. 3H,I), similar to *Itgb4* loss (Fig. 2D). As when *Itgb4* was targeted, K10/K14 (Krt14, keratin 14) dual-positive basal cells in *Lama3¹⁴³⁶* epidermis were enriched in the RFP⁺ population compared to RFP⁻ (Fig. 3D,E), demonstrating that increased delamination contributes to the high differentiation index. Thus, basal layer retention is mediated by integrin-β4, through its ligand, laminin-α3.

To shed additional light on the mechanisms regulating delamination, we revisited an intriguing finding that aberrant Notch signaling leads to reduced HDs and loss of integrin-β4 at the DEJ (Blanpain et al., 2006). While Notch is known to promote epidermal differentiation (Blanpain et al., 2006; Williams et al., 2011; Moriyama et al., 2008), whether it impacts delamination specifically is not known. To test this, we used both gain- and loss-of-function models: (1) *Krt14^Cre;Rosa^NICD1-IRES-GFP* transgenics overexpressing the Notch1 intracellular domain (hereafter referred to as *Rosa^NICD*), and (2) *Rbpj^fl/fl* embryos transduced with lentiviral Cre-RFP (*Rbpj* cKO).

First, we investigated whether altering Notch activity impacted integrin-β4 expression levels at the DEJ. Indeed, integrin-β4 was nearly undetectable in *Rosa^NICD* embryos, while levels were increased >2-fold in *Rbpj* cKOs (Fig. 4A-D). This is consistent with Notch acting upstream of, and negatively regulating, integrin-β4, and suggests that Notch activity may precede delamination. To test this, we used two independent Notch reporters – a transgenic mouse (Duncan et al., 2005) and a lentiviral construct (Williams et al., 2011) – to visualize

**DEVELOPMENT**

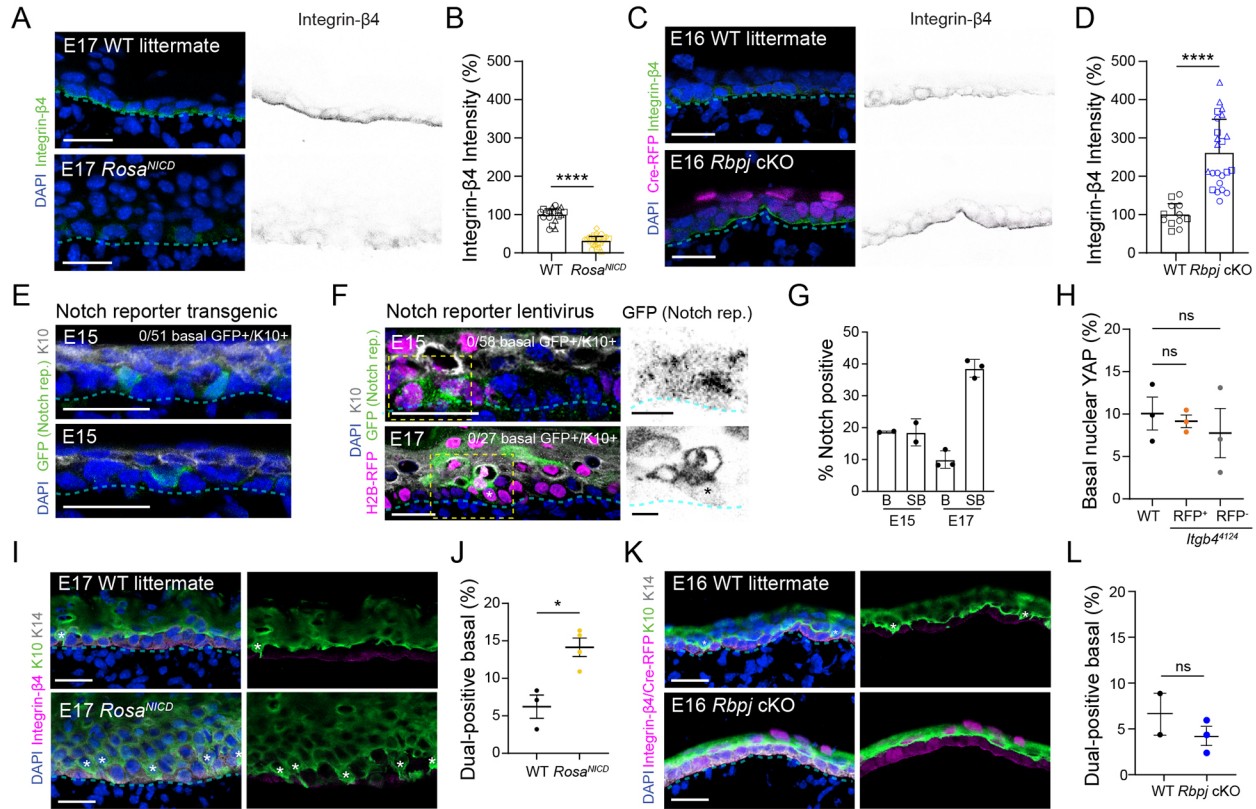

**Fig. 4. Notch signaling regulates integrin-β4 levels and delamination.** (A-D) Confocal images of E17 Cre-negative control versus *Rosa^NICD* epidermis (A) and E16 WT littermate versus *Rbpj* cKO epidermis (C), with single channel images of integrin-β4 on the right, and associated quantification of fluorescence intensity (B,D); *n*=3 except for WT control in *Rbpj* cohort (*n*=2). (E-G) Images of Notch reporter (NR) transgenic (E) and LUGGIGE NR-transduced epidermis (F), and associated quantification (G). NR shown in green; RFP marks cells transduced with reporter. (H) Quantification of nuclear YAP in *Itgb4^4124* epidermis. (I,K) Confocal images of E17 Cre-negative control versus *Rosa^NICD* epidermis (I) and E16 WT littermate versus *Rbpj* cKO epidermis (K); asterisks indicate dual-positive cells. (J,L) Quantification of dual-positive cells in WT versus mutant. Each dot represents a biological replicate in G,H,J,L, or FOV in B, D, where shapes designate litters. Basement membrane is indicated with cyan dashed line. Scale bars: 25 µm (A,C,E,F,I,K); 10 µm (F, insets). ns, not significant; *P<0.05; ****P<0.0001 (*t*-test).

Notch activity during epidermal stratification (Fig. 4E,F). While most Notch reporter-positive (NR⁺) cells were found suprabasally at E17, NR⁺ cells were also observed in the basal layer, particularly at E15 (Fig. 4G). Out of the 136 total basal NR⁺ cells observed, not one was K10⁺, suggesting that Notch activation is an early, and perhaps initiating, step in the delamination process. While YAP/TAZ activity has been shown to be reduced in JEB cells, and YAP can negatively regulate Notch in the epidermis (De Rosa et al., 2019; Totaro et al., 2017), we did not observe any significant difference in nuclear YAP expression upon *Itgb4* knockdown (Fig. 4H).

We previously showed that *Rbpj* loss does not affect ACDs (Williams et al., 2011). Similarly, no change in division orientation was observed in *Rosa^NICD* embryos, despite abnormal thickening of the basal layer and increased suprabasal mitoses (Fig. S5A,B), suggesting that the role of Notch in differentiation may be through delamination. Quantification of K10/K14 dual-positive basal cells revealed a significant increase in delamination upon NICD1 overexpression, and a trend toward decreased delamination in *Rbpj* cKOs (Fig. 4I-K). Collectively, these data suggest that Notch activation promotes differentiation specifically by delamination, via reduction of HD adhesions.

Classically, delamination has been viewed as the major driver of epidermal differentiation (Watt, 1984; Watt and Green, 1982), but in recent years it has become clear that both ACDs and delamination play crucial, but context-dependent, roles (Smart, 1970; Williams et al., 2011). While perpendicular divisions are a key driver of

differentiation during peak stratification, they are not required for initial stratification or adult homeostasis in skin, although ACDs persist into adulthood in other stratified epithelia (Byrd et al., 2019; Ipponjima et al., 2016; Lechler and Fuchs, 2005; Rompolas et al., 2016; Williams et al., 2011). Here, we show that cell–matrix adhesions also play a minor role in regulating telophase correction, adding to the list of extrinsic factors such as tissue geometry, apoptosis and cell–cell adhesions that influence division orientation (Box et al., 2019; Lough et al., 2019; Soffer et al., 2022).

Although direct evidence of embryonic delamination is limited, as we show here (Fig. 2K), it appears to occur on a timescale of minutes to hours, while in the adult, this process can take days (Cockburn et al., 2022; Damen et al., 2021; Mesa et al., 2018). Delaminating cells have been likened to less-fit 'losers' in a competition between basal cells; however, while loser cells delaminate during peak stratification, they die by apoptosis at earlier stages (Ellis et al., 2019). At a molecular level, the desmosomal adhesion protein Dsg1 has been linked to delamination *in vitro* by reducing tension at E-cadherin (cadherin 1) cell–cell junctions (Nekrasova et al., 2018), while another study concluded that increased cell–cell adhesion via E-cadherin promotes delamination (Miroshnikova et al., 2018). These seemingly conflicting results could potentially be explained by the fact that the first study used immature (24 h) raft cultures while the second used mature, fully stratified (6 day) cultures, hinting at potential differences between early and peak delamination mechanisms.

DEVELOPMENT

Finally, our data indicate that the hyper-differentiative phenotype induced by NICD1 can be partially explained by an increase in delamination propensity, while Notch loss in *Rbpj* mutants enhances integrin-β4 expression and decreases delamination, albeit not significantly. Mechanistically, we feel it is unlikely that HD components are direct Notch targets, because Notch generally functions as a transcriptional activator rather than repressor; however, the Notch target Hes1 can function as a transcriptional repressor (Moriyama et al., 2008). Moreover, although YAP/TAZ has been proposed as a mechanistic link between cell–matrix adhesions, Notch signaling and differentiation (De Rosa et al., 2019; Totaro et al., 2017), we found no evidence that *Itgb4* loss impacts YAP activity (Fig. 4H). Nonetheless, we provide the first evidence for the role of integrin-β4/laminin-332 adhesion in basal cell retention and differentiation and reveal mechanistic insights into how Notch signaling influences delamination.

## MATERIALS AND METHODS
### Animals
All mice in this study were housed in the AAALAC-accredited (000329), USDA registered (55-R-004) and NIH welfare-assured [D16-00256 (A3410-01)] animal facility at the University of North Carolina at Chapel Hill. All procedures and surgeries were conducted under IACUC-approved animal protocols (19-155, 22-121, 25-086). CD1 WT outbred mice (Charles River #022) were used for all fixed tissue imaging of lentiviral-infected embryos. For live explant imaging of the epidermis, mice on a mixed C57BL76/J background with alleles for *Krt14Cre; Rosa26*$^{mTmG}$ {*Rosa26mTmG* [*Gt(ROSA)26Sortm4(ACTB-tdTomato,-EGFP)Luo/J*]; The Jackson Laboratory stock #007576} animals were used. For lineage-tracing experiments, *Krt14Cre*$^{ERT2}$*; Rosa26*$^{mTmG}$ [*Krt14Cre*$^{ERT2}$*; Tg(KRT14-cre/ERT)20Efu*; The Jackson Laboratory stock #005107] animals were used, and were treated with a single low dose (200 µl of 20 mg/ml by gavage) of tamoxifen at E15.5 before harvesting at E17.5. Transgenic Notch reporter animals were obtained from The Jackson Laboratory [*Tg(Cp-EGFP)25Gaia/J*, stock #005854]. *Rosa*$^{NICD-IRES-GFP}$ mice were obtained from The Jackson Laboratory [strain *Gt(ROSA)26Sortm1(Notch1)Dam/J*; stock #008159]. Conditional *Rbpj*$^{fl/fl}$ mice (Tanigaki et al., 2002) were obtained as previously described (Williams et al., 2011). All animals were infected with constructs described below using an *in utero* lentiviral injection. Note that all ages were defined as E0.5 the morning a vaginal plug was discovered. Additional developmental aging was performed by measuring the crown-rump length of embryos using a Fuji Visualsonics Vevo 2100 ultrasound. Note that animals on the C57BL6/J background typically grew at a slower rate, roughly 0.5 days developmentally behind the CD1 strain mice.

### Lentiviral injections
The lentiviral injection procedures were performed according to Beronja et al. (2010), under the approved, aforementioned protocols (see 'Animals' section). Briefly, pregnant mice were anesthetized with 3% isoflurane for no more than 1 h and analgesics provided (5 mg/kg meloxicam and 1-4 mg/kg bupivacaine). One uterine horn was isolated, and two or three developing embryos were revealed at a time through a membrane, into a dish of sterile 1× PBS. Lentivirus (1.0-1.5 µl) was delivered using a Drummond microinjector into the amniotic cavity, visualized by ultrasound. There to six embryos were injected per pregnant dam, per surgery. Uninfected animals within the same litter were used as WT littermate controls. Following surgery, all embryos were returned into the body cavity of the dam, and surgical site was closed with sterile sutures and surgical staples. Mice were monitored for 4-7 days following surgery, and harvested at the appropriate developmental age, between E14.5 and E17.5.

### Constructs and RNAi
shRNAs targeting *Itgb4* or *Lama3* were generated according to previously reported cloning protocols, described by Beronja et al. (2010). NCBI accession numbers for a given shRNA clone are identified by gene name with the last four digits of the 21-nucleotide target sequence, for example

*Itgb4*$^{4124}$. shRNA constructs were cloned into vector backbones containing either the H2B-RFP reporter sequence (*in vivo*), or a puromycin-selectable sequence (*in vitro*). Both versions of the shRNA were packaged into lentivirus by using HEK 293FT cells, pMD2.G (Addgene plasmid #12259) and psPAX2 (Addgene plasmid #12260) helper plasmids. Some modification to prior published methods were made. First, plasmids were transfected with Lipofectamine 3000 rather than calcium chloride, which yielded higher transfection efficiencies. To scale the Lipofectamine reagents effectively, 358 µl of Lipofectamine was added to 8.642 ml of OptiMEM (for a total of 9 ml). The 9 ml of DNA mix and p3000 reagent were then added to two 500 cm$^2$ plates containing 81 ml of OptiMEM media each. Second, OptiMEM viral production media (discontinued by the manufacturer) was replaced with DMEM containing 5% fetal bovine serum and 2 µg/ml insulin. Other supplements (1% L-glut, 1% sodium bicarbonate, 1% pen-strep, 1% sodium pyruvate) remained unchanged.

### shRNA validation
Primary cultured keratinocytes were transduced with an shRNA-containing pLKO.1-based lentivirus construct containing the puromycin-resistance gene. Forty-eight hours following infection, cells were transferred into E low Ca$^{2+}$ media (Nowak and Fuchs, 2009) supplemented with 2 µg/ml puromycin and incubated for at least 7 days, with at least one passage during that time course. Selected cell lines were then lysed using the RNeasy Mini Kit (QIAGEN) to isolate RNA. cDNA was generated using the SuperScript IV VILO Kit (Life Technologies, 11756050). RT-qPCR was then performed on dilute cDNA to determine mRNA knockdown efficiency using the TaqMan Fast Advance System on a Thermo Fisher QuantStudio 6 Pro. The following TaqMan Probes were used: *Ppib* (Mm00478295_m1) and *Itgb4* (Mm01266840_m1). Cyclophilin B (*Ppib*) was used as the house keeping gene, and 'Scramble' shRNA puro-selected cell lines were used as the reference control.

### TEM
To analyze epidermal ultrastructure, embryo heads from WT littermate controls and *Itgb4*$^{4124}$-transduced animals were harvested and immediately placed into fresh, prewarmed TEM fixative (4% paraformaldehyde, 2.5% glutaraldehyde, 0.05 M sodium cacodylate, pH 7.4, 5 mM calcium chloride and 1 mM magnesium chloride) and incubated for 2 h at room temperature. A total of three mutant and three controls embryos were submitted for processing, with two each at E15.5 and one at E17.5. Heads were kept at 4°C for no more than 3 weeks, until sample delivery to the UNC Microscopy Services Laboratory Electron Microscopy Team. Nape skin was removed in small square pieces, resin embedded, sagittally sectioned and placed on TEM grids. Nape skin was collected as that is typically the most highly transduced area of epidermis. Since TEM without immunogold cannot distinguish between transduced and non-transduced cells, this approach maximized the likelihood that the majority of cells would be RFP$^+$. High transduction efficiency was also validated by immunofluorescence on matched backskin samples. During TEM imaging, we first identified the DEJ and captured reference images at 1500× and 8000× magnification for reference and to characterize gross architecture. Edges of the sample were not analyzed. At least 25 sequential images were taken at 30,000× magnification at the DEJ. Representative images of HDs from the DEJ and desmosomes from the lateral membranes were taken at 50,000-60,000× magnification. Desmosomes across conditions appeared normal. HDs were characterized as 'mature' or 'immature' as follows: mature HDs are roughly triangular, electron dense and contain anchoring fibrils extending into the lamina densa and clear keratin filament attachments intracellularly, while immature HDs lack one or more of these defining characteristics (Guo et al., 1995).

### Antibodies, immunohistochemistry and fixed imaging
*Itgb4*$^{4124}$-, *Itgb4*$^{2326}$-, *Lama3*$^{1436}$- and *Rbpj*$^{fl/fl}$-infected animals and their WT littermate controls were embedded whole into cassettes with Optimal Cutting Temperature (OCT, TissueTek) to avoid any damage to the presumed fragile epidermis. Scramble-infected and NICD epidermis were skinned and flattened onto Whatman paper before embedding. All tissue was then fresh-frozen at −20°C and stored at −80°C for at least 24 h prior to tissue sectioning. Knockdown/knockout animals were always mounted

together with at least one WT littermate to allow for direct comparison within the same slide for all immunohistochemistry. Frozen samples were then sectioned (10-12 μm thick) on a Leica CM1950 cryostat. Staining of sectioned animals was conducted as previously described by Lough et al. (2019). Images were acquired using LAS AF software on a Leica TCS SPE-II, 4-laser confocal system on a DM5500 upright microscope. ACS Apochromat 40×/1.5 NA oil objectives were used for all fixed imaging. Pinhole was set at 1.0, with gain, offset and laser power adjusted to maximize LUT range and minimize photobleaching.

The following antibodies were used: survivin (rabbit mAb 71G4B7, Cell Signaling Technology, 2808S, AB_2063948; 1:500), LGN (rabbit, Millipore ABT174, AB_2916327; 1:2000), phospho-histone H3 (rat, Abcam ab10543, AB_2295065; 1:1000), cleaved-caspase 3 (rabbit, Cell Signaling Technology 9661, AB_2341188; 1:1000), integrin-β4 (rat, Thermo Fisher Scientific 553745, AB_395027; 1:1000), cytokeratin-14 (chicken, BioLegend 906004, AB_2616962; 1:1000), cytokeratin-10 (rabbit, BioLegend 905401, AB_2565049; 1:1000), GFP (chicken, Abcam ab13970, AB_300798; 1:1000), mCherry (rat, Life Technologies M11217, AB_2536611; 1:1000), RFP (rabbit, MBL PM005, AB_591279; 1:1000), laminin-5/laminin-β3 (rabbit, Abcam, ab97765, AB_10678844; 1:500), YAP (rabbit, Cell Signaling Technology 14074, AB_2650491; 1:300), integrin-α6/CD49f (rat, Bio-Rad MCA2034A647 AB_324801; 1:50), integrin-β1/CD29 (hamster, BioLegend 102202 AB_312879; 1:100).

The following secondaries all raised in donkey (with exceptions noted), and were diluted in gelatin block with normal donkey serum to reduce off target antibody staining: anti-rabbit Alexa Fluor 488 (Invitrogen, A32766TR; 1:1000), anti-rabbit Rhodamine Red-X (Jackson ImmunoResearch Laboratories, 711-295-152; 1:500), anti-rabbit Cy5 (Jackson ImmunoResearch Laboratories, 711-175-152; 1:400), anti-rat Alexa Fluor 488 (Invitrogen, A48269TR; 1:1000), anti-rat Rhodamine Red-X (Jackson ImmunoResearch Laboratories, 712-295-153; 1:500), anti-rat Cy5 (Jackson ImmunoResearch Laboratories, 712-175-153; 1:400), goat anti-guinea pig Alexa Fluor 488 (Invitrogen, A-11073; 1:1000), anti-guinea pig Rhodamine Red-X (Jackson ImmunoResearch Laboratories, 706-295-148; 1:500), anti-guinea pig Cy5 (Jackson ImmunoResearch Laboratories, 706-175-148; 1:400), anti-goat Alexa Fluor 488 (Invitrogen, A-11055; 1:1000), anti-goat Cy5 (Jackson ImmunoResearch Laboratories, 705-175-147; 1:400), anti-mouse IgG Alexa Fluor 488 (Invitrogen, A-21202; 1:1000), anti-mouse IgG Cy5 (Jackson ImmunoResearch Laboratories, 715-175-151; 1:400) and anti-mouse IgM Cy3 (Jackson ImmunoResearch Laboratories, 715-295-140; 1:500).

### EdU injection and analysis
To determine the number of cycling basal cells, two independent dams were injected intraperitoneally with 10 μl/g dam weight of 5 mM EdU in $H_2O$ following lentiviral surgery, 2 h before harvest. Secondary detection of EdU in tissue sections was performed by incubating tissue with ClickIt Chemistry in PBS containing 2 mM $CuSO_4$, 8 μM sulfo-cyanine5 azide, 20 mg/ml ascorbate for 30 min following secondary antibody treatment. EdU-positive basal cells were then counted, and the percentage of EdU⁺/H2B-RFP⁺ dual-positive cells was determined relative to total H2B-RFP⁺ basal cells, and likewise for RFP⁻ and WT cells.

### Live imaging
Live imaging was performed as previously described (Descovich et al., 2023; Lough et al., 2019; Cetera et al., 2018). In brief, epidermal explants at E16.5 were harvested from the mid-back of infected or uninfected $Krt14^{Cre}$; $Rosa26^{mTmG}$ animals. Dissected explants were sandwiched between a 1% Agar/F-Media patty (Ham's F-12 Media, DMEM, 5% fetal bovine serum, sodium bicarbonate, sodium pyruvate and Pen/Step L-glutamine) and a gas-permeable membrane dish (Lumox 3000). Imaging was performed using either a Zeiss LSM-900 confocal or Andor Dragonfly Leica SP-5 spinning disk confocal microscope. On both microscopes, samples were maintained in a humidified, 37°C temperature-controlled chamber, with 5% $CO_2$, and 20×/0.80 Plan Apo and 40×/1.4 Oil Plan Apo objectives were used to image samples on the LSM-900. A Zyla Plus 4.2MP sCMOS camera and a HC PL APO 20×/0.75 LWD objective were used for imaging explants on the Dragonfly. For experiments on both microscopes, a 20-30 μm z-stack was

acquired every 5 min for between 2-6 h. Imaging close to the epidermal edge was avoided due to edge disorganization and damage, limiting the effect a wound response could have on basal cell behavior. Imaging files were then deconvoluted using AutoQuantX 3.1, individual image files of dividing cells were generated using the TimeSeriesAnalyzer v2.0, and division angles were measured in Fiji. Measurements of division orientation were taken at anaphase onset (t=0 min) and anaphase completion (t=60 min).

### Data analysis
#### HD quantification
HDs were identified in 30,000× magnification images. Density was quantified as the number of HDs per μm length of tissue imaged. Sequential 30,000× images had adjacent, but not overlapping, views of the DEJ. Mature HDs were categorized based on the following criteria: (1) electron density, (2) ∼10 nm in width, (3) presence of clear anchoring fibrils, and (4) visible keratin filament extensions into the cytoplasm. Immature HDs were identified as being electron dense but lacking one or more of the other identifiable features of mature HDs. Immature HDs also did cause noticeable depressions or flexing in the basal membrane of the basal keratinocytes, like mature HDs do.

#### Division orientation
Division orientation measurements were conducted on fixed and live-imaged samples according to previously published methods (Descovich et al., 2023; Lough et al., 2019; Byrd et al., 2016; Williams et al., 2014, 2011). In short, survivin was used as a midbody marker of dividing cells in telophase (late mitotic marker). The angle between pairs of daughter nuclei (labeled with DAPI, identified with survivin) was measured as a vector between the center of the DAPI signals and parallel to the basement membrane (integrin-β4). In local areas of integrin-β4 depletion due to knockdown, the basement membrane angle was determined using residual integrin-β4 staining and the basal surface signal of keratin 14. Cells in hair placodes were excluded. Small z-stacks were taken from dividing cells to obtain the best imaging plane. All division angle vectors were calculated in Fiji. Live imaging division angles were measured in a similar fashion from individual frames over time, or at specific time points (t=0 and t=60). In RFP⁻ cells, with no H2B-RFP nuclear signal, nuclear positioning was estimated based on cell volume/membrane shape changes.

#### DEJ antibody intensity
Basement membrane intensity of integrin-β4, integrin-α6 and laminin-β3 protein localization and intensity was quantified using a line profile analysis. *Itgb4* knockdown and control tissue were sectioned and stained on the same slides. Epidermal sections were stained for anti-integrin-β4 detected with Alexa Fluor 488, anti-mCherry with Rhodamine Red X and anti-K14 with Cy5. On the same day, using the same imaging settings, several fields of view (FOV) were taken from anterior to posterior of the mouse. In Fiji, 10-pixel-thick lines were drawn according to the shape of the basal surface of cytokeratin 14 staining corresponding to either RFP⁺ zones, RFP⁻ zones (at least three cells in a row), and in zones from WT animals. The sum fluorescence intensity of the measured region was then divided by the total length of the region measured. The intensity per RFP⁺, RFP⁻ and WT zone were normalized to the average of the WT zone for that corresponding WT littermate, such that the average of WT intensity was 1.0 (100%). Data are plotted as the individual regions measured, where each shape corresponds to a biological replicate for that genotype. At least three biological replicates, across multiple FOV, were measured for each intensity measurement. A similar method was used to analyze integrin-β1 basal cell intensity, but a 50-pixel-thick line was used to measure the entirety of the basal layer.

#### LGN and pericentrin localization
Localization of LGN crescents was categorically determined from pHH3-positive basal cells blind to RFP status. Chi-square analyses were used to determine the difference in count across the categories for RFP⁺, RFP⁻ and WT basal cells. Pericentrin localization was determined by measuring the angle between the pericentrin puncta, bisecting the nucleus, relative to/parallel to the basement membrane. In areas of integrin-β4 local depletion, keratin 14 or laminin-β3 signal was used to determine the basal surface of

the tissue. Angles from individual cells across multiple animals were represented as a scatter plot with angles ranging from 0-90° from the basement membrane.

## Spinous layer thickness

Spinous layer thickness was determined from knockdown and control tissue sectioned and stained on the same slides. Tissue was stained for anti-cytokeratin 10 detected with Alexa Fluor 488, anti-mCherry and anti-integrin β4 with Rhodamine Red X, and anti-cytokeratin 14 with Cy5. All sections were sampled with 1 μm optical sections for a total volume of 15 μm using the 40× oil objective of the Leica SPE II DM5500 microscope. Each epidermis was imaged along the anterior-posterior axis, to achieve multiple FOV per embryo. Infected animals were imaged in areas of relatively high basal cell infection. Images were then processed in Fiji using a macro with the following operations: (1) generate maximum projection of only the K10 channel, (2) threshold using Huang metrics, (3) 'fill holes' of the mask, (4) use 'open' function to eliminate background pixel contribution, (5) select and measure mask area. Mask TIFFs were then traced along the basal surface to determine the length of the epidermis measured. Total area of the spinous layer was then normalized to the length of the epidermis measured. Data are represented as the average of the multiple positions along the anterior-posterior axis per animal. Embryos aged E16.5 and E17.5 were pooled per genotype such that each genotype is represented by at least five embryos.

## K10/K14 dual-positivity frequency calculations in basal cells

To determine the rate of dual positivity in fixed samples, epidermal sections were stained and imaged in the same way as the spinous layer thickness quantification described above. At least six FOV were taken from anterior to posterior. Using the Fiji plugin 'Cell Counter', six categories of cells were determined: (1) total basal cells (K14 and DAPI), (2) dual positive (K14, K10 and DAPI), (3) total infected basal cells (K14, H2BRFP and DAPI), (4) dual-positive cells with clear basement membrane contact (K14, K10, DAPI, full or reduced contact, i.e. balloon shaped), (5) total spinous layer cells (K10, DAPI), and (6) infected spinous layer cells (K10, H2BRFP and DAPI). Then, totals for each category per animal were determined and represented as a fraction over that total population: RFP$^-$ dual positive with contact/total RFP$^-$ basal cells versus RFP$^+$ delamination percentage=RFP$^+$ dual positive with contact/total RFP$^+$ basal cells. Each dot represents the total percentage of dual-positive basal cells per genotype for an animal, where at least 25 basal cells (either RFP$^+$ or RFP$^-$) were counted for each animal. This threshold was intended to reduce the effect size of dual-positive events over a small population. In animals with either low basal cell infection (low RFP$^+$ counts) or high basal cell infection (low RFP$^-$ cell counts), more technical replicates were analyzed to reach a minimum of 25 cells.

## Differentiation index

Based on the cell count categories from the percentage of dual-positive basal cells per FOV, we can determine the percentage of spinous cells that are H2B-RFP positive relative to the percentage of basal cells that are H2B-RFP positive. H2B-RFP positioning serves as a lineage-tracing tool because cells are infected when the epidermis is a single layer of keratinocytes (all basal resident). Suprabasal to basal (SB:B) ratios were calculated based on the ratio of the percentage of spinous RFP$^+$ over the percentage of basal RFP$^+$ cells. An SB:B ratio of 1.0 corresponds with equal infection rates in each layer, indicating no occupancy preference. SB:B ratios higher than 1.0 indicate a pro-differentiative fate in that FOV, while SB:B ratios below 1.0 indicate a pro-basal resident fate in that FOV. In Fig. 2B,C, the SB:B ratios per FOV are plotted, where the x-axis indicates the percentage basal infection, and the y-axis indicates the percentage suprabasal infection. Closed circles represent positions with a basal infection below 50%, well below the threshold for saturating tissue infection. Open circles represent SB:B ratios where the basal infection is above 50%.

## Statistical analyses and graphs

All measurements were recorded into Microsoft Excel worksheets and exported to GraphPad Prism v10, where graphs and statistical analyses were performed. Error bars represent s.e.m. Data were first analyzed for normality using the Anderson–Darling, Shapiro–Wilk and Kolmogorov–Smirnov tests.

For data that did not pass all three tests for normality, a Mann–Whitney test (non-parametric, comparison of rank) was performed to determine statistical significance. For data that passed all three tests for normality, Welch's t-test (parametric, comparison of means) was performed to determine statistical significance. Cumulative frequency distributions of division orientation angles (Fig. 2F; Fig. S3A) were analyzed using a non-parametric, Kolmgorov–Smirnov test; no statistically significant differences were found between groups. Statistical tests of significance for categorical data [telophase correction outcomes (Fig. 2G) and lineage tracing (Fig. 3A-C)] were performed by an observed versus expected distribution analysis, where expected values were determined by the distribution of the internal control. For comparisons of the percentage of dual-positive basal cells (Figs 2D and 3I), individual t-tests were performed to compare WT versus the RFP$^+$ means. Note that all dual-positive basal frequencies passed tests for normality, and thus were analyzed by Welch's unpaired, parametric, t-tests. Comparisons of Itgb4 and Lama3 knockdown RFP$^+$, dual-positive basal cell rates to Scramble RFP$^+$, dual-positive basal cell rates were also statistically significant by Welch's t-test (Itgb4$^{4124}$ versus Scramble *P=0.0126; Itgb4$^{2326}$ versus Scramble **P=0.0012; Lama3$^{1436}$ versus Scramble *P=0.0121). Figures were generated first in Prism and uploaded to Adobe Illustrator as .eps files. Sample sizes and cell numbers were chosen based on the power analyses conducted for similar studies (Descovich et al., 2023; Lough et al., 2019; Williams et al., 2014).

## Acknowledgements

We would like to acknowledge Williams lab members Amber Altrieth, Lauren Griffith, Carlos Patino Descovich and Bethany Brown for their insightful feedback, technical support and manuscript editing. Joint laboratory meeting discussions between the Peifer, Bergstralh-Finegan and Lovegrove labs have also been instrumental in the organization of the manuscript. We thank members of the Devenport Lab, specifically Danelle and Brandon Trejo, for their support with PS-Multiview imaging and feedback. We thank members of the John Morris IV lab for the use of their YAP antibody. We are immensely grateful to Pablo Ariel, Kristen White and Jillann Madren, and Kathleen Clardy in the Microscopy Services Laboratory (MSL) core facility for light microscopy assistance and TEM tissue preparation and imaging. The Dragonfly spinning disk microscope was supported by NIH grant S10 OD030223.

## Competing interests

The authors declare no competing or financial interests.

## Author contributions

Conceptualization: J.S.K., K.J.L., S.E.W.; Data curation: J.S.K., S.E.W.; Formal analysis: J.S.K., S.E.W.; Funding acquisition: J.S.K., S.E.W.; Investigation: J.S.K., K.J.L., S.E.W.; Methodology: J.S.K., K.J.L., S.E.W.; Project administration: S.E.W.; Resources: J.S.K., S.E.W.; Software: J.S.K., S.E.W.; Supervision: J.S.K., S.E.W.; Validation: J.S.K., S.E.W.; Visualization: J.S.K., S.E.W.; Writing – original draft: J.S.K., S.E.W.; Writing – review & editing: J.S.K., K.J.L., S.E.W.

## Funding

This work was supported by the National Institutes of Health (R01 AR077591 to S.E.W.; F31 DE033915-01 to J.S.K.), the United States-Israel Binational Science Foundation (2019230 to S.E.W.); and the University of North Carolina at Chapel Hill Cell Biology and Physiology Department (T32 GM133364 to J.S.K.). Open Access funding provided by University of North Carolina at Chapel Hill. Deposited in PMC for immediate release.

## Data and resource availability

All relevant data and details of resources can be found within the article and its supplementary information.

## Peer review history

The peer review history is available online at https://journals.biologists.com/dev/lookup/doi/10.1242/dev.205210.reviewer-comments.pdf

## Special Issue

This article is part of the Special Issue 'The Extracellular Environment in Development, Regeneration and Stem Cells', edited by Alex Hughes and Rashmi Priya. See related articles at https://journals.biologists.com/dev/issue/153/16.

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
