## [Peer Review File · Development (Cambridge, England)]

Hemidesmosomes and Notch signaling regulate epidermal differentiation via delamination

Juliet Stanton King, Kendall J. Lough and Scott E. Williams
DOI: 10.1242/dev.205210

Editor: Liz Robertson

Review timeline

Original submission:	3 September 2025
Editorial decision:	20 October 2025
First revision received:	23 February 2026
Accepted:	18 March 2026

Original submission

First decision letter

MS ID#: dev.205210

MS TITLE: Hemidesmosomes regulate epidermal differentiation during embryogenesis

AUTHORS: Juliet Stanton King, Kendall J. Lough and Scott E. Williams

Dear Dr Williams,

I have now received all the referees' reports on the above manuscript, and have reached a decision. The referees' comments are appended below, or you can access them online: please go to:

As you will see, the referees express considerable interest in your work, but have some significant criticisms and recommend a substantial revision of your manuscript before we can consider publication. If you are able to revise the manuscript along the lines suggested, which may involve further experiments, I will be happy receive a revised version of the manuscript. Your revised paper will be re-reviewed by one or more of the original referees, and acceptance of your manuscript will depend on your addressing satisfactorily the reviewers' major concerns. Please also note that Development will normally permit only one round of major revision. If it would be helpful, you are welcome to contact us to discuss your revision in greater detail. Please send us a point-by-point response indicating your plans for addressing the referees' comments, and we will look over this and provide further guidance.

Please attend to all of the reviewers' comments and ensure that you upload both a 'clean' version of your Word file, along with a highlighted version clearly showing where you have made changes in the revised manuscript. Please avoid using 'Tracked changes' in Word files as these are lost in PDF conversion. I should be grateful if you would also provide a point-by-point response detailing how you have dealt with the points raised by the reviewers in the 'Response to Reviewers' box. If you do not agree with any of their criticisms or suggestions please explain clearly why this is so.

Reviewer 1

Advance summary and potential significance to field

This short report by King et al investigates the function of hemidesmosomes in skin development using in utero lentiviral knockdown to generate mosaic epidermal loss of Integrin-B4 (Itgb4) and its

ligand laminin- $\alpha 3\beta 3\gamma 2$ (Lama3). The authors show that Integrin- $\beta 4$ knockdown in the epidermis does not lead to epidermal-dermal separation as it does in adult skin, but rather promotes telophase correction of oblique cell divisions and promotes the retention of cells in the basal layer. They further show that hyperactive Notch signaling, known to drive excessive stratification, decreases *Itgb4* and elevates delamination. Overall, the study is well designed and controlled, and the data are clearly presented and carefully quantified. The conclusions throughout the report are generally well supported by the images and data that they have presented.

These findings are significant because they define a role for hemidesmosome components in the embryonic stages of epidermal differentiation and stratification that goes beyond simply attachment to the extracellular matrix. These findings suggest that basal progenitor cells sense their position and regulate their fate through signaling associated with hemidesmosomes. This paper should be of broad interest to researchers interested in cell adhesion, tissue architecture, division orientation and differentiation and skin biology.

Comments for the author

Addressing the following points would help clarify the findings and strengthen the conclusions:

Major Comments:

* One of the main conclusions of the paper is that "hemidesmosome adhesions regulate differentiation through delamination". As the data do not directly visualize cells delaminating, but infer they must be delaminating based on K10 expression and only minor changes in division orientation. I feel the language should be toned down - cells are likely delaminating when hemidesmosomes are depleted, or when NICD is expressed, but the evidence is indirect.

* It would be helpful if the authors could discuss the relationship between *Itgb4* expression, differentiation marker expression and physical delamination from the basal layer in terms of the sequence of events. Loss of hemidesmosomes leads to differentiation, while differentiation (NICD) decreases hemidesmosomes. This seems useful for the cell to sense its level of attachment to make a fate decision, and also be able to make a fate decision that alters its attachment. Some discussion of this relationship in terms of building the epidermis would be a nice addition to the discussion.

Other comments:

* Figure 1C. RFP-negative cells show reduced *Itgb4* intensity, suggesting there is a non-cell autonomous effect, which conflicts with the description in the text.

* Figure 1E. Data is pooled from E16.5 and E17.5 stages, but these are very different in terms of epidermal maturation. Data should be separated by stage.

* Figure 2. shRNA 2326 appears to have a stronger SB:B defect, but the authors use the other hairpin for most experiments throughout the paper. Could the authors include a statement about their reasoning? It's possible the data shown represent partial loss-of-function phenotypes.

* Figure 3: For the division orientation data, it would be helpful to know the final positions (SB or B), or fate, of the daughter cells. For example, what is the fate of daughters for cells in the 'stay oblique' category.

* Figure 4. It is claimed that the elevated number of K10/K14+ dual-positive delaminating cells are likely the primary contributor to hyper differentiation in NICD expressing skin, but this is not fully supported as there is an increase in suprabasal mitoses, some of which are dividing oblique and perpendicular, which was surprising to me.

* Figure 3G: I found the layout confusing and was unsure how the different panels relate to one another. I suggest daughter cell correlated color fills/outlines, inclusion of a wild type control panel, and addition of supplemental movies for clarity.

* Unclear throughout the paper what developmental stage each figure shows, so this should be stated in the figure legends if not the figure themselves.

* N should be at least 3 throughout; adjust and report embryo numbers.

* Unclear in some cases which data are live vs fixed.

Minor:

* Figure 1D: I'm not sure what this is showing, I think this can be removed without losing any meaning.

* Figure 1G/H: Needs representative E17.5 image to match quantification.

- * Figure 4C: EM data. Are both images knockdown? There should be a control comparison either to Fig 1 or to a WT example.
- * Figure 4I. Should the WT controls be separated into basal vs suprabasal? There isn't much explanation of this panel in the main text and the oblique and perpendicular divisions in the SB layer are somewhat surprising.
- * Supp. Figure 2F: Apoptosis data. I didn't understand the legend next to the Itgb4 datapoints. Also, the two populations appear to be statistically different but the stats don't bear this out. The meaning of the y axis is also unclear. There needs to be more detail on the statistics and quantification in the legend and writing for this figure.
- * There is no Supplemental Fig. 3.
- * Supplemental Fig. 4C: n=1 for these data? Is this the number of replicates for the laminin experiments the main figure? Ideally three biological replicates would be included.
- * Supp. Figure 4D: needs a control reference image

Reviewer 2

Advance summary and potential significance to field

This manuscript by King et al. demonstrate that hemidesmosomes in embryonic epidermis restrict delamination and promote telophase correction during stratification to prevent premature basal cell differentiation. Using an in utero mosaic knockdown approach targeting Itgb4 or Lama3, the authors show that these processes are regulated autonomously by basal keratinocytes in vivo, independent from neighboring cells and external stimuli cause by blistering. The authors further provide strong evidence connecting Notch signaling and hemidesmosomes to delamination and basal cell retention, which are essential for proper skin development and regeneration. Overall, this work offers new mechanistic insights into how adhesive complexes integrate with differentiation pathways during embryogenesis and will make a valuable contribution to the field if some areas can be clarified and strengthened to fully convey the significance of the findings.

Comments for the author

1. The manuscript shows that Itgb4 knockdown cells are more likely to remain oblique after telophase. However, the downstream consequences of these persistent oblique divisions are not clear. Do these cells subsequently express differentiation markers, similar to delaminated cells?
2. Figure 2 indicates that proliferation of basal cells is not affected after Itgb4 knockdown, but whether suprabasal mitoses are increased has not been examined. Quantification of EdU-positive suprabasal cells would help rule out the possibility that loss of integrin and telophase correction induces suprabasal proliferation, which could contribute to the increased suprabasal transduction.
3. If laminin is reduced, what happens to integrin- β 4 levels at the DEJ? Quantification of Itgb4 localization in Lama3 knockdown tissue would help clarify whether the phenotype arises from loss of ligand binding, destabilization of integrin- β 4, or both.
4. The study tries to link Notch signaling, hemidesmosomes, and delamination, but the directionality is unclear. Would blocking Notch increase integrin levels and reduce delamination frequency? Addressing this would strengthen the mechanistic model.
5. Some graphs treat each image as an independent data point rather than averaging at the animal level. For robustness, a minimum of three biological replicates (animals) should be used for each condition. Please clarify this in the Methods section or in figure legends.
6. It would be helpful to show a WT image alongside the Itgb4 knockdown images in Figures 4C and S4D for direct comparison.
7. Error bars are missing from all cumulative bar plots. Please add them to convey statistical reliability.
8. In figure 3G and S2C, the asterisks are not explained in figure legend. Please revise the legends.
9. There is no Fig. S3, please correct the numbering of Figure S4.
10. In Figure 2D, Itgb4 knockdown skin also shows greater variation toward the right side of the line, reflecting a higher basal transduction rate. Please address and discuss the observation in the main text.

Reviewer 3*Advance summary and potential significance to field*

Loss of either $\alpha 6$ or $\beta 4$ subunits or the ligand laminin331 results in skin blistering diseases that can be very severe, and similar, loss of either $\alpha 6$ or $\beta 4$ or lam332 in mice also results in severe skin blistering. This paper assesses the role of the $\alpha 6\beta 4$ integrin and its ligand in regulation spindle orientation and differentiation during development in the absence of overt blistering using in utero lentiviral knockdown in the embryonic epidermis to obtain a mosaic cells with reduced $\alpha 6\beta 4$ integrin. Overall, the authors uncover a small but significant change in spindle orientation, with more oblique spindles and reduced perpendicular spindles. Moreover, they find that $\beta 4$ KD cells are unable to induce the telophase correction pathway previously described by this group. However, differences here are small, and in part based on small numbers, and not significant.

As the authors acknowledge these observations likely do not explain why more RFP+ cells representing $\beta 4$ KD cells end up suprabasally compared to control KD RFP+ cells. They find that loss of either $\beta 4$ or the A3 chain of Lama332 in the absence of overt blistering promotes delamination and premature differentiation, in agreement with findings on human patient keratinocytes that show premature differentiation, as well as by other findings that found that loss of Collagen XVII, a further hemidesmosomal component, also promotes premature differentiation.

Finally, the authors show that increasing Notch signaling in basal that is known to promote differentiation not unexpectedly seems to quite significantly downregulate $\beta 4$ expression that seems much worse than any of the KD experiments, still the percentage of double positive K14/K10 cells is similar to $\beta 4$ KD, suggesting that delamination may not be a direct consequence of simple less $\beta 4$.

Overall, the data describe the effects of reduced $\alpha 6\beta 4$ on spindle orientation and cell positioning as well as on delamination as measured by double positive K14/K10 cells but effects are relatively small. It remains unclear what, if any, the developmental consequences are of these small changes in either delamination or spindle orientation, as it seems not to alter thickness, proliferation etc., and other papers also did not find any overt changes also not during development in stratification and differentiation (e.g., DiPersio et al., JCS 2000, see other citations used by author in the paper). Overall, the new knowledge and insights are therefore limited and some of the data is not entirely convincing.

Comments for the author

1. Hemidesmosomes are poorly characterized, which is necessary as the EM images really show quite severely impaired or very very immature hemidesmosomes if one can still name them like that. What happens to other hemidesmosomal components? There is very little characterization, in human and in mice $\alpha 6$ will associate with $\beta 1$, where is $\alpha 6$ localized? What happens to laminin332 and collagen XVII? It seems that the change in lamina densa suggest that there is some very mild micro-blisters. Is there an initial upregulation of $\beta 1$ integrin?

2. In some cases their analysis are based on rather small numbers. For the initial change in spindle orientation (Fig.2 F and G) the authors put cells in 3 categories using only 43 cells for the $\beta 4$ KD whereas for the control 116 cells are used. Indeed, the observed shift of perpendicular to more planar is not significant. Similar, in figure 3, the changes in E and F on which the authors base their conclusions that $\beta 4$ regulates telophase rescue pathway, are also very small (5 versus 9 cells in E and 8 and 16 cells in F).

3. The quality of some images is not great, 2F, and H especially. In fig.2E survivin staining is of poor quality, it is not entirely clear from the image whether the wt indeed shows a perpendicular division. In figure 3B and G some of the pseudocoloring actually makes it hard to see whether these cells are indeed dividing even if I do understand that the imaging resolution in life embryo imaging is not optimal.

4. the authors use K10/K14 as a measure of delamination based on previous data, but can the authors rule out that the double positive cells increase due to the changes seen in division orientation with these double positive cells being the result of the oblique divisions that are not

corrected with one of the daughter cells sort of sit in between basal and suprabasal expressing both markers, and not so much a result of increased delamination?

5. It would strengthen the data if the authors should quantify the extend of BM contact of the double positive cells to show that these cells have less contact and are delaminating as some of the cells shown in fig.4A that are K10 positive seem to have as much contact as wt cells. Why are the suprabasal cells not clearly K10 positive in Fig.4A?

6. The authors do not discuss previous findings that might be very relevant for the increase in differentiation they observe. Human patient keratinocytes mutated for either lam332 or ab4 show increased differentiation in vitro and in vivo that has been linked to reduced Yap (De Rosa et al., Cell Reports, 2019), which then itself may promote Notch signaling (Totaro et al., Nature Com, 2017). Is Yap changed under mosaic conditions and may this explain changes in delamination?

First revision

Author response to reviewers' comments

We greatly appreciate the time and attention the editorial team and reviewers took to revise and strengthen our manuscript, which we have retitled, “**Hemidesmosomes and Notch signaling regulate epidermal differentiation via delamination**”. We were pleased to read the constructive and generally favorable comments from the panel of reviewers. For example, **Reviewer 1** stated that “*these findings are significant*” and “*should be of broad interest.*” **Reviewer 1** goes on to say: “*Overall, the study is well designed and controlled, and the data are clearly presented and carefully quantified. The conclusions throughout the report are generally well supported by the images and data that they have presented.*” **Reviewer 2** summarizes by concluding: “*Overall, this work offers new mechanistic insights into how adhesive complexes integrate with differentiation pathways during embryogenesis and will make a valuable contribution to the field...*”

Below is a summarized response to each of the reviewer comments. When reviewer comments were similar or addressed a common them, they were combined, but they have been color-coded as follows: **Reviewer 1**, **Reviewer 2**, **Reviewer 3**.

Language and narrative

One of the main conclusions of the paper is that “hemidesmosome adhesions regulate differentiation through delamination”. As the data do not directly visualize cells delaminating, but infer they must be delaminating based on K10 expression and only minor changes in division orientation. I feel the language should be toned down - cells are likely delaminating when hemidesmosomes are depleted, or when NICD is expressed, but the evidence is indirect.

We agree with this statement and have made several textual changes to soften our language, as well as being explicit in references to the interpretation of certain assays (e.g., we refer to the K10/K14 assay as “dual-positive basal cells” rather than “delaminating cells”). However, we have now added several direct lines of evidence to support the conclusion that integrin-B4 regulates delamination behavior. The most significant is the addition of clonal lineage tracing in Fig. 3A-C. We performed LUGGIGE with *Itgb4*⁴¹²⁴ shRNA on the *K14*^{CreER}; *Rosa*^{mTmG} background, and administered a single dose of tamoxifen at E15.5 to induce recombination and clonal labeling, then harvested 2 days later. We scored 51 RFP+ and 72 RFP- clones and found a significant increase of delamination events in the RFP+ population over the RFP- population. While this method is a great way to analyze terminal fate of mutant cells, it is labor intensive and relies on a bit of luck to obtain embryos that are the correct genotype AND successfully injected. Therefore, we believe that analyzing rates of K10+ in the basal layer in fixed tissue on the CD1 background is higher throughput and therefore better suited for analyzing multiple genotypes and ages.

Second, we now provide direct evidence that delamination can be imaged live *ex vivo* by implementing a novel approach developed by the Devenport lab called planar-sagittal (PS) MultiView (Jones et al, *JCB*, 2023). This approach uses epidermal explants laid over a “cliff” on an agarose patty so that the imaging plane shifts from xy (*en face*) to xz (sagittal) (Fig. 2H). Importantly, this allows imaging far from the edge of the explant, thus avoiding any potential “wound” effects. We now show movie stills from a cell undergoing delamination (Fig. 2K), which reveal how it gradually loses contact with the basement membrane over a ~30 minute period. Notably, the morphology of this cell looks quite similar to dual- positive basal cells in fixed tissue. We attempted several times to perform PS MultiView imaging on explants that were transduced with the *Itgb4*⁴¹²⁴ H2B-RFP virus, but ran into several technical challenges. First, the field of view that can be imaged with PS MultiView is much smaller than a typical *en face* view, so the odds of encountering a delaminating cell in any given field of view are low. There are also fewer potential fields that can be multiplexed, and z-drift is a bigger issue with this imaging format. Moreover, the RFP signal was difficult to detect on both microscopes we tried (Andor Spinning Disc and Zeiss LSM910 confocal), making it challenging to be sure whether a cell was RFP+ or RFP-. Finally, to be able to make any comparisons about the frequency of delamination between RFP+ and RFP- cells we would need to capture a large number (we estimated 20 by power analyses) to draw any meaningful conclusions. For now this remains an exciting but low-throughput methodology, so we were unable to apply it to our dream experiment of capturing *Itgb4* knockdown cells delaminating.

Figure 1C. RFP-negative cells show reduced *Itgb4* intensity, suggesting there is a non-cell autonomous effect, which conflicts with the description in the text.

We agree with the reviewer and have therefore removed the language relating to cell autonomy. That said, there is clearly a stronger effect for both *Itgb4* hairpins in the RFP+ population compared to the RFP- population. Several possible explanations could account for the lower integrin-β4 protein levels observed in the *Itgb4*⁴¹²⁴ hairpin: 1) some RFP- cells could be “false-negatives” and actually express the shRNA, even if we are unable to detect nuclear H2B-RFP, 2) the approach we used focused on mosaic, rather than highly-transduced areas. In mosaic areas, it is much more likely that residual protein from neighboring RFP- cells (some of which could be out of the plane of imaging) could be observed near RFP+ cells.

Figure 2. shRNA 2326 appears to have a stronger SB:B defect, but the authors use the other hairpin for most experiments throughout the paper. Could the authors include a statement about their reasoning? It's possible the data shown represent partial loss-of-function phenotypes.

We agree with this argument about the possibility of a partial loss-of-function phenotype in the *Itgb4*²³²⁶ hairpin, which by our *in vitro* and *in vivo* quantification, appears to be the weaker one. This was, in fact, the primary reason we used the “stronger” *Itgb4*⁴¹²⁴ hairpin for most of our studies. After bolstering our “n values” for the dual-positive basal cell experiments, we now see that the *Itgb4*⁴¹²⁴ shRNA has a stronger effect in the RFP+ cells than the *Itgb4*²³²⁶ shRNA (Fig. 3E). While the effect on the “differentiation index” (SB:B) ratio remains stronger in the *Itgb4*²³²⁶ shRNA (Fig. 2D), there are several potential explanations. First, as the reviewer suggests, partial loss-of-function could have a stronger phenotype than stronger loss-of-function. Second, the differentiation index is not a measure of delamination, because ACDs and differences in proliferation rates could also contribute to this index, so there may be other behaviors caused by the *Itgb4*²³²⁶ that we were unable to detect which could impact basal residency. Third, the titer of the *Itgb4*²³²⁶ virus was weaker than the *Itgb4*⁴¹²⁴ virus, leading to lower overall basal transduction levels. We believe that since this is a “clonal” assay, it is more accurate at lower densities, so the *Itgb4*⁴¹²⁴ tissue could be closer to approaching saturation. Unfortunately, a lengthy discussion of this in the text, when limited to 3000 words, is not possible.

We have now included further discussion of the RFP- integrin-β4 intensity changes in both the primary hairpin (*Itgb4*⁴¹²⁴) and the secondary hairpin (*Itgb4*²³²⁶). The differences in integrin-β4 intensity in RFP- regions between the two hairpins could also explain the observed

differences in SB:B ratio phenotypes between the two hairpins. This is further explained in the following point.

In figure 3G and S2C, the asterisks are not explained in figure legend. Please revise the legends.

Thank you for noticing this oversight. The asterisks in Fig. 3G (now Fig. S3H) indicate the position of the basal daughter. The asterisk in S2C indicates EdU+ basal cells. These explanations have been added to the legends.

In Figure 2D, *Itgb4* knockdown skin also shows greater variation toward the right side of the line, reflecting a higher basal transduction rate. Please address and discuss the observation in the main text.

As described above, the “clonal” power of the assay is lost at higher transduction rates. Therefore, we chose to compare the SB:B ratio of *Itgb4* mutants versus *Scramble* at basal infection rates at or below 50%, which are the data we now display in Fig. 2D. In a tissue where there are roughly similar numbers of RFP+ and RFP- basal cells, changes in differentiative fate are easier to observe. It is not surprising that *Scramble* mutant cells show a roughly 1:1 SB:B ratio at low and high basal infection rates, because RFP+ and RFP- cells have no competitive advantage relative to each other. On the other hand, when RFP+ cells are at a competitive disadvantage (as it appears to be for *Itgb4* knockdowns), this becomes more apparent when they are surrounded by a greater number of WT cells.

The authors do not discuss previous findings that might be very relevant for the increase in differentiation they observe. Human patient keratinocytes mutated for either lam332 or ab4 show increased differentiation in vitro and in vivo that has been linked to reduced Yap (De Rosa et al., Cell Reports, 2019), which then itself may promote Notch signaling (Totaro et al., Nature Com, 2017). Is Yap changed under mosaic conditions and may this explain changes in delamination?

We are grateful to the reviewer for bringing these studies to our attention. We now cite these two papers in the main text and discussion and have performed additional experiments to investigate the relationship between integrin-B4 loss and YAP. Specifically, we have stained 3 *Itgb4*⁴¹²⁴ mutant animals from 3 litters with anti-YAP (D8H1X) XP Rabbit mAb. We then quantified rates of nuclear YAP in the RFP+ basal cells compared to WT and RFP-, and displayed these data in Fig. 4H. We find a slight, but insignificant decrease in nuclear YAP in *Itgb4* knockdown cells compared to uninfected cells. However, nuclear YAP is but one readout of YAP signaling, and while we would love to investigate this further, we believe it is beyond the scope for the present study.

Notch Signaling and Differentiation Mechanisms

It would be helpful if the authors could discuss the relationship between *Itgb4* expression, differentiation marker expression and physical delamination from the basal layer in terms of the sequence of events. Loss of hemidesmosomes leads to differentiation, while differentiation (NICD) decreases hemidesmosomes. This seems useful for the cell to sense its level of attachment to make a fate decision, and also be able to make a fate decision that alters its attachment. Some discussion of this relationship in terms of building the epidermis would be a nice addition to the discussion.

The study tries to link Notch signaling, hemidesmosomes, and delamination, but the directionality is unclear. Would blocking Notch increase integrin levels and reduce delamination frequency? Addressing this would strengthen the mechanistic model.

We thank the reviewers for their interest in the link between integrin-B4 expression, delamination and Notch signaling. We have performed several new lines of experiments, which have been added to new Figure 4, which focuses exclusively on Notch by utilizing loss- and gain-of-function, as well as a Notch reporter. First, we find that endogenous Notch activation can be observed in basal keratinocytes in early stratifying epidermis (Fig. 4E-G). While this was

observed by us previously (Williams, *NCB*, 2014; Fig. S2a,b), we have now investigated this more rigorously. We examined two different Notch reporter models (transgenic and lentivirus) at two different ages (E15 and E17). Importantly, across 136 Notch Reporter+ basal keratinocytes, none were Keratin-10 positive. This data suggests that transient Notch activation occurs prior to Keratin-10 expression and progressive loss of contact with the basement membrane. Second, we analyzed epidermis from conditional *Rbpj* knockout embryos, which serves as an excellent loss-of-function model since the transcriptional co-factor Rbpj is required for canonical Notch signaling. We find integrin- β 4 levels are increased in these mutants, and that there is a trend toward decreased dual-positive basal cells (Fig. 4C,D,K,L). We have also now quantified integrin- β 4 levels in the NICD1 mutant and show that they are significantly decreased (Fig. 4B). Together with our previous data showing that NICD1 overexpression increases dual-positive basal cells, these data now strongly support a model that Notch promotes (and perhaps initiates) epidermal delamination.

Fate of oblique divisions

The manuscript shows that *Itgb4* knockdown cells are more likely to remain oblique after telophase. However, the downstream consequences of these persistent oblique divisions are not clear. Do these cells subsequently express differentiation markers, similar to delaminated cells?

Figure 3: For the division orientation data, it would be helpful to know the final positions (SB or B), or fate, of the daughter cells. For example, what is the fate of daughters for cells in the 'stay oblique' category.

The authors use K10/K14 as a measure of delamination based on previous data, but can the authors rule out that the double positive cells increase due to the changes seen in division orientation with these double positive cells being the result of the oblique divisions that are not corrected with one of the daughter cells sort of sit in between basal and suprabasal expressing both markers, and not so much a result of increased delamination?

We find the potential relationship between failed telophase correction and delamination interesting, too. In reanalyzing our live data, we find that following persistent oblique divisions through live imaging, oblique daughters eventually take on a more squamous or flat shape (greater than 60mins post anaphase). This indicates that persistent oblique daughters are fated to differentiate and do not maintain progenitor function. Furthermore, data from our lab has previously shown that mutants with dysfunctional telophase correction have a thickened Keratin-10 layer and suprabasal cell density, suggesting that increased oblique divisions tend to have a differentiative fate (Lough et al. 2019).

Finally our new lineage tracing data (Fig. 3A-C) shows little change in the proportion of planar/symmetric and perpendicular/asymmetric divisions between *Itgb4*⁴¹²⁴ RFP+ and RFP-clones, suggesting that the effect of *Itgb4* on oriented cell divisions, whether in initial spindle positioning or telophase correction, is mild. We have since de-emphasized this finding in the text and focused more on the delamination aspect.

Implication of suprabasal mitoses in *Itgb4* mutants

Figure 4I. Should the WT controls be separated into basal vs suprabasal? There isn't much explanation of this panel in the main text and the oblique and perpendicular divisions in the SB layer are somewhat surprising. Figure 2 indicates that proliferation of basal cells is not affected after *Itgb4* knockdown, but whether suprabasal mitoses are increased has not been examined. Quantification of EdU-positive suprabasal cells would help rule out the possibility that loss of integrin and telophase correction induces suprabasal proliferation, which could contribute to the increased suprabasal transduction.

We did not see signs of suprabasal divisions in the *Itgb4* mutants in our Survivin or PHH3 data, as those would have been noted. We would argue that analysis of either Survivin or PHH3 stained epidermis in the mutant backgrounds would be a better way to analyze suprabasal mitoses as the presence of EdU+ suprabasal cells does not rule out that a cell delaminated following entry into S- phase. The NICD1 mutant is the only example where we observed an increase in

suprabasal mitoses, which is why we presented these data separately in the cumulative frequency histogram of division orientation (Fig. S4A).

Layout

There is no Supplemental Fig. 3.

There is no Fig. S3, please correct the numbering of Figure S4.

Thank you for noticing this oversight. In the revised version of the manuscript there are now 5 supplemental figures, labeled S1-S5. S1 pertains to Fig. 1; S3 and S3 to Fig. 2, S4 to Fig. 3 and S5 to Fig. 4.

Unclear throughout the paper what developmental stage each figure shows, so this should be stated in the figure legends if not the figure themselves.

This has now been explicitly stated in the figure panels and legends. In the one figure where ages are pooled (E16.5 and E17.5), the data points associated with each age have been indicated with different symbols (Fig. 1D)

Figure 3G: I found the layout confusing and was unsure how the different panel relate to one another. I suggest daughter cell correlated color fills/outlines, inclusion of a wild type control panel, and addition of supplemental movies for clarity.

In figure 3B and G some of the pseudocoloring actually makes it hard to see whether these cells are indeed dividing even if I do understand that the imaging resolution in live embryo imaging is not optimal.

We have inverted the live imaging representative images to increase the membrane contrast so that the pseudocoloring is not masking the cell shapes. We hope that this will improve the clarity. We also moved the *en face* views to the Supplement (Fig. 3H)

Unclear in some cases which data are live vs fixed.

We have added additional titles on the figures to clarify the difference.

Figure 1D: I'm not sure what this is showing, I think this can be removed without losing any meaning. Thank you. We thought it looked cool but realize it has limited and redundant value. This panel has been removed.

Figure 1G/H: Needs representative E17.5 image to match quantification.

These images are at E17.5, which we now state clearly in the legend. We elected not show an E15.5 image because there are very few hemidesmosomes present.

Supp. Figure 4D: needs a control reference image

This figure panel has been removed due to limited space.

Error bars are missing from all cumulative bar plots. Please add them to convey statistical reliability.

Thank you for your attention to detail. To clarify, the cumulative bar graphs are representing the (cumulative) proportion of total cells with a certain division angle. Therefore, there cannot be error bars.

However, statistical analyses can be performed by Kolmogorov-Smirnov tests, which we have done and included in the legends and text. Even after adding additional data to boost the “n” values for the *Itgb4*⁴¹²⁴ shRNA, there is no statistically significant difference between the RFP+ and RFP- (or WT littermate) distributions.

The quality of some images is not great, 2F, and H especially. In fig.2E Survivin staining is of

poor quality, it is not entirely clear from the image whether the wt indeed shows a perpendicular division.

Thank you for the feedback. We have chosen some new, and we believe better, representative images for the Survivin staining in Fig. 2E. We have moved the LGN localization data to the Supplement (Fig. S3B), due to space, and because it is negative data. After years of working with this antibody, I can say that it does not always yield the “greatest” images.

Why are the suprabasal cells not clearly K10 positive in Fig.4A?

This is one of the vagaries of this antibody, which becomes more pronounced at older ages as the suprabasal layers become wider and more undulated. We suspect this could be due to “ruffling” of the section as it adheres to the slide. Nonetheless, we have selected other images for this panel (now Fig. 3D), which look a bit better. Note that these are now max projection images which can counter some of the out-of-plane-of-focus issues.

Increase N and clarify age/genotype

N should be at least 3 throughout; adjust and report embryo numbers.

Thank you for your attention to our rigor and reproducibility. We have performed additional experiments to increase the number of biological replicates for all studies so that they are at least three per genotype per age, and usually greater, with a few exceptions: 1) the TEM data, and 2) the E17.5 timepoint for the *Itgb4*²³²⁶ shRNA (Fig. 1D), 3) E15.5 lenti Notch reporter (Fig. 4G) and 4) WT littermates for *Rbpj* cKO (Fig. 4L). This increased rigor is most apparent in Fig. 3E (dual-positive quantification) where we have doubled the number of *Lama3* knockdowns from 3 to 6 and added two more biological replicates for Scramble and each of the two *Itgb4* shRNAs. Moreover, we have adjusted our graphs such that each biological replicate within a given genotype is indicated with a different shape data point (e.g., Fig. 1C,J,L; Fig. 3G,I; Fig. 4B,D), with multiple data points analyzed for each animal in a genotype. We have also adjusted our figure legends to accurately reflect the replicates. We

* Supplemental Fig. 4C: n=1 for these data? Is this the number of replicates for the laminin experiments the main figure? Ideally three biological replicates would be included.

We have increased the number of *Lama3*¹⁴³⁶ mutants such that there are 3 mutants from E16.5 and 3 from E17.5.

It would be helpful to show a WT image alongside the *Itgb4* knockdown images in Figures 4C and S4D for direct comparison.

We have added WT comparisons for these images, now shown in Fig. S4B.

Some graphs treat each image as an independent data point rather than averaging at the animal level. For robustness, a minimum of three biological replicates (animals) should be used for each condition. Please clarify this in the Methods section or in figure legends.

As noted above, we now make it very clear when multiple data points come from a biological replicate by utilizing different shapes to plot the data points. As mentioned earlier, with few exceptions, the number of biological replicates now equals or exceeds three per condition per age.

In some cases their analysis are based on rather small numbers. For the initial change in spindle orientation (Fig.2 F and G) the authors put cells in 3 categories using only 43 cells for the b4 KD whereas for the control 116 cells are used. Indeed, the observed shift of perpendicular to more planar is not significant. Similar, in figure 3, the changes in E and F on which the authors base their conclusions that b4 regulates telophase rescue pathway, are also very small (5 versus 9 cells in E and 8 and 16 cells in F).

We have performed additional experiments to increase the number of cells analyzed for our division orientation data. We have increased the number of *Itgb4*⁴¹²⁴ RFP+ cells from 43 to 65; RFP- cells from 83 to 135 and WT littermate cells from 116 to 125. We also moved the RFP- data into the main figure (Fig. 2F). With this additional rigor, there now appears to be very little difference among distributions, other than an increase in obliques (not significant) in the RFP+ group. We recognize that we have low N for the live division orientation data, though these come from several imaging sessions of 3 biological replicates, and we did analyze a total of ~100 cells. The n of the initial oblique group is necessarily small because this is a relatively small fraction of the total.

Protein Abundance at the Dermal-Epidermal Junction

If laminin is reduced, what happens to integrin- β 4 levels at the DEJ? Quantification of *Itgb4* localization in *Lama3* knockdown tissue would help clarify whether the phenotype arises from loss of ligand binding, destabilization of integrin- β 4, or both.

Hemidesmosomes are poorly characterized, which is necessary as the EM images really show quite severely impaired or very very immature hemidesmosomes if one can still name them like that. What happens to other hemidesmosomal components? There is very little characterization, in human and in mice α 6 will associate with β 1, where is α 6 localized? What happens to laminin332 and collagen XVII? It seems that the change in lamina densa suggest that there is some very mild micro-blisters. Is there an initial upregulation of β 1 integrin?

To address these comments, we have performed additional analyses of quantifications of components of the DEJ. We now show that integrin- α 6, which dimerizes with integrin- β 4, is also significantly reduced in *Itgb4*⁴¹²⁴ RFP+ regions (Fig 1I,J). While we initially believed that laminin- β 3 and integrin- β 1 may have also been altered, upon robust quantification across multiple biological replicates, we now find no significant differences between *Itgb4*⁴¹²⁴ RFP+ regions and littermate controls (Fig. 1K,L; S1G,H). While it would have been nice to examine integrin expression in *Lama3*¹⁴³⁶ mutants as well, these embryos display a very high suprabasal enrichment score, and subsequently lack large regions of RFP+ cells in the basal layer. For this reason, while we do not see a striking difference in integrin- β 4 intensity at the DEJ in *Lama3*¹⁴³⁶ mutant cells, we have not quantified the intensity.

It would strengthen the data if the authors should quantify the extend of BM contact of the double positive cells to show that these cells have less contact and are delaminating as some of the cells shown in fig.4A that are K10 positive seem to have as much contact as wt cells.

This is a fantastic idea and one we would like to pursue, but believe that this would require super- resolution imaging to accomplish. At the level of light microscopy, we do not see any significant difference in integrin- β 4 intensity in these dual-positive cells, which could be explained by several possibilities: 1) as mentioned, we are may be unable to detect subtle differences with this approach, 2) it may be a matter of surface rather than total integrin- β 4, and we cannot distinguish between them with this method, 3) the reduction in integrin- β 4 may be transient and/or precede expression of K10. Evidence in support of this latter possibility is provided by the fact that we never observed a Notch Reporter+ K10+ basal cell. Thus, if Notch is indeed the initial signal to delaminate, integrin downregulation could precede K10 expression, and the double-positive cells we are catching are those in the last stages of delamination.

Second decision letter

MS ID#: dev.205210R1

MS TITLE: Hemidesmosomes and Notch signaling regulate epidermal differentiation via delamination

AUTHORS: Juliet Stanton King, Kendall J. Lough and Scott E. Williams

Dear Dr Williams,

I am happy to tell you that your manuscript has been accepted for publication in Development, pending our standard publication integrity checks.

Reviewer 1

Advance summary and potential significance to field

The authors have done an excellent job of responding to the reviewers' comments and critiques. The addition of new live imaging data, Notch loss-of-function experiments and Notch signaling reporters paint a fuller picture of how hemidesmosomes act to retain cells in the basal layer and how Notch-mediated differentiation events downregulate hemidesmosome components to promote basal layer exit.

Reviewer 2

Advance summary and potential significance to field

Comments for the author

The authors have carefully addressed the concerns raised in the previous review. The revisions have improved the clarity and rigor of the manuscript, and I have no further major concerns.